# Risk and rates of hospitalisation in young children: A prospective study of a South African birth cohort

Catherine J. Wedderburn[1,2,3]*, Julia Bondar[1,4], Marilyn T. Lake[1], Raymond Nhapi[1], Whitney Barnett[5], Mark P. Nicol[6], Liz Goddard[1], Heather J. Zar[1]

1 Department of Paediatrics and Child Health, Red Cross War Memorial Children's Hospital and SA Medical Research Council Unit on Child and Adolescent Health, University of Cape Town, Cape Town, South Africa, 2 Department of Clinical Research, London School of Hygiene & Tropical Medicine, London, United Kingdom, 3 Neuroscience Institute, University of Cape Town, Cape Town, South Africa, 4 School of Medicine, Universidade Federal do Rio Grande do Sul, Porto Alegre, Brazil, 5 Department of Psychology and Human Development, Vanderbilt University, Nashville, TN, United States of America, 6 Marshall Centre, School of Biomedical Sciences, University of Western Australia, Perth, Australia

* catherine.wedderburn@uct.ac.za

**Data Availability Statement:** Data cannot be shared publicly because of ethical conditions with which study investigators are obliged to comply. Access to the project data is restricted to

## Abstract

Children in sub-Saharan Africa (SSA) are disproportionately affected by morbidity and mortality. There is also a growing vulnerable population of children who are HIV-exposed uninfected (HEU). Understanding reasons and risk factors for early-life child hospitalisation will help optimise interventions to improve health outcomes. We investigated hospitalisations from birth to two years in a South African birth cohort study. Mother-child pairs in the Drakenstein Child Health Study were followed from birth to two years with active surveillance for hospital admission and investigation of aetiology and outcome. Incidence, duration, cause, and factors associated with child hospitalisation were investigated, and compared between HEU and HIV-unexposed uninfected (HUU) children. Of 1136 children (247 HEU; 889 HUU), 314 (28%) children were hospitalised in 430 episodes despite >98% childhood vaccination coverage. The highest hospitalisation rate was from 0–6 months, decreasing thereafter; 20% (84/430) of hospitalisations occurred in neonates at birth. Amongst hospitalisations subsequent to discharge after birth, 83% (288/346) had an infectious cause; lower respiratory tract infection (LRTI) was the most common cause (49%;169/346) with respiratory syncytial virus (RSV) responsible for 31% of LRTIs; from 0–6 months, RSV-LRTI accounted for 22% (36/164) of all-cause hospitalisations. HIV exposure was associated with increased incidence rates of hospitalisation in infants (IRR 1.63 [95% CI 1.29–2.05]) and longer hospital admission (p = 0.004). Prematurity (HR 2.82 [95% CI 2.28–3.49]), delayed infant vaccinations (HR 1.43 [95% CI 1.12–1.82]), or raised maternal HIV viral load in HEU infants were risk factors for hospitalisation; breastfeeding was protective (HR 0.69 [95% CI 0.53–0.90]). In conclusion, children in SSA experience high rates of hospitalisation in early life. Infectious causes, especially RSV-LRTI, underly most hospital admissions. HEU children are at greater risk of hospitalisation in infancy compared to HUU children. Available strategies such as promoting breastfeeding, timely vaccination, and optimising antenatal maternal HIV

nominated investigators approved by the University of Cape Town Human Research Ethics Committee, as per the consent document. Interested, qualified researchers may request to access this data by contacting the Drakenstein Child Health Study (via lesley.workman@uct.ac.za) to submit a formal data use request and ensure required ethical approval received prior to use.

**Funding:** The study was funded by the Bill and Melinda Gates Foundation [OPP 1017641]; additional support from the National Institutes of Health (H3Africa 1U01AI110466-01A1), the National Research Foundation, South Africa and the MRC South Africa, extramural unit. CJW was supported by the Wellcome Trust through a Research Training Fellowship [203525/Z/16/Z]. HZ is supported by the SA-MRC. MPN is funded by an Australian National Health and Medical Research Council Investigator Award APP1174455. The funders had no role in study design, data collection and analysis, decision to publish, or preparation of the manuscript.

**Competing interests:** The authors have declared that no competing interests exist.

care should be strengthened. New interventions to prevent RSV may have additional impact in reducing hospitalisation.

## Introduction

The Sustainable Development Goals highlight the need to prevent child morbidity and mortality, with the aim of reducing under-five mortality to less than 25 deaths per 1,000 live births by 2030 [1]. Children in low and middle-income countries (LMIC) are disproportionately affected by morbidity and mortality in early life [2], particularly in sub-Saharan Africa (SSA) [3]. Hospital admission represents a key marker of child health, vulnerability, and risk of mortality worldwide [4–6], which may be used to define intervention priorities [7]. However, there are limited recent data on rates and causes of hospitalisations in LMICs, especially in the context of HIV exposure. Quantifying hospitalisation burden, identifying causes and determinants of hospital admission, as well as populations who are most at-risk, will allow development of targeted prevention and intervention strategies [2, 8].

In SSA, HIV is a key risk factor for morbidity and mortality. Prevalence of HIV infection in pregnant women continues to remain high in SSA, however, access to antiretroviral therapy (ART) in pregnancy has expanded and vertical transmission of HIV has decreased substantially [9]. Therefore, numbers of children who are HIV-exposed and uninfected (HEU) are growing, with this population representing over 20% of live births in some countries [9, 10]. Several studies have shown that, when compared to HIV-unexposed and uninfected children (HUU), children who are HEU are at higher risk of mortality [11, 12], impaired growth [13, 14], neurocognitive impairment [15], and severe infections [16–18]. Studies have also reported that early-life hospitalisation may be increased in children who are HEU compared to HUU [19–21]. However, more evidence is needed to understand the causes and determinants of hospitalisation to improve child health outcomes.

Investigating the factors that impact child morbidity and hospitalisation in SSA is critical to inform and optimise interventions [22], and accelerate progress towards the Sustainable Development Goals. We prospectively investigated hospitalisation rates and determinants amongst children from birth through two years of life in the Drakenstein Child Health Study (DCHS), a population-based birth cohort outside Cape Town, South Africa, in an area of high HIV prevalence.

## Methods

### Study population

Pregnant women were enrolled in the DCHS from March 5, 2012 to March 31, 2015 from two primary public healthcare facilities in Paarl, Western Cape, South Africa. This is a low socio-economic peri-urban setting with high rates of infection and psychosocial risk factors [23]. The area has a well-established public healthcare system which provides antenatal and postnatal child health services, including 23 primary health clinics and Paarl Regional Hospital serving a catchment population of approximately 200,000 people. All women who were over the age of 18 years and intended to be resident in the area for at least one year were eligible [23]. Women gave informed consent and received standard antenatal care. All births occurred at the single central hospital, Paarl Regional Hospital, serving this area. Mother-child pairs were followed from birth through two years [23, 24]. Children received vaccination according to the South African Extended Programme on Immunisation schedule which included the 13-valent

pneumococcal vaccine, rotavirus vaccine [25], and vitamin A supplementation as per national policy. Delay in vaccination was defined at 14 weeks and 9 months as any vaccinations delayed by more than 2 weeks from the due date.

## Hospitalisations

Hospitalisations were tracked by active surveillance at Paarl Regional Hospital, the presenting and only referral public hospital for all children in this area. Study staff conducted daily rounds at the emergency unit and inpatient ward of Paarl Regional Hospital, to ensure real time surveillance. Additionally, hospital staff and mothers enrolled were also sensitised to contact study staff when an admission occurred; there was a 24 hour study phone manned by a member of the study team available to all mothers [26]. Detailed information regarding each admission episode including sociodemographic factors, anthropometry, diagnosis and treatment were obtained by study staff and abstracted from hospital folders, with data recorded on study questionnaires. The main causes of hospitalisations were documented in study-specific standard proformas and grouped into lower respiratory tract infection (LRTI); gastroenteritis; other infections; and other causes (for example accident and trauma) based on the treating clinician diagnosis. Two study personnel with clinical training independently quality checked the proformas and assigned the cause groupings. Where multiple causes were recorded, the most severe cause was used in this analysis as defined by the reviewers as being likely to have led to hospitalisation. The diagnosis of LRTI was made according to WHO clinical case definitions; all children had a nasopharyngeal sample taken at the time of LRTI for potential pathogens including respiratory syncytial virus (RSV) using nucleic acid amplification testing, as described previously [27, 28].

## Measures

**HIV.** Maternal HIV status was established antenatally at enrolment. As per South African prevention of mother-to-child transmission (PMTCT) guidelines, HIV testing of pregnant and breastfeeding women was performed routinely every 3 months during pregnancy and the postnatal period. All mothers with HIV were enrolled in the national PMTCT programme and initiated on ART as recommended by the Western Cape guidelines at the time. Enrolment started in March 2012 and ended in March 2015, Before May 2013, ART guidelines followed "Option A" which was based on a pregnant woman's clinical and immunological status, providing ART for life or zidovudine (AZT) starting at 14 weeks gestation. In 2013, South Africa adopted Option B+, with the recommendation to initiate triple ART for all pregnant women for life (efavirenz, emtricitabine and tenofovir) [29, 30]. HIV-exposed children were tested for HIV at 6–10 weeks of age using a PCR test, 9 months and 18 months using a rapid test, ELISA or PCR. HIV data were collected by examining clinic and hospital folder information [31]. Given the low vertical transmission rate in this area, the two children with HIV infection were excluded from this analysis as there was insufficient power to detect an effect in this population. Clinical measures of maternal CD4 and plasma viral load measurements antenatally were obtained through the National Health Laboratory Service system. Where there was more than one result the highest viral load and lowest CD4 were included, as a marker of maternal HIV disease severity.

**Breastfeeding.** Detailed longitudinal data were collected on type of feeding from birth, breastfeeding practices, formula feeding and introduction and use of complimentary foods based on maternal report at each study visit (6–10 weeks, 14 weeks, 6 months, 9 months, 12 months, 18 months and 24 months after birth). Where conflicting responses were given, earlier reports were prioritised; they were considered more accurate due to being closer in time to the

event reported, which should minimise recall bias. Duration of exclusive breastfeeding was derived based on maternal report and defined as occurring until the first maternal report of introduction of solid food or formula for child [32]. Dichotomous variables were also included based on international and national recommendations for duration of exclusive breastfeeding and any breastfeeding.

**Sociodemographic variables.**　Sociodemographic variables were measured using validated questionnaires administered by trained study staff at an antenatal visit at 28 to 32 weeks of gestation. Sociodemographic variables including maternal education, household factors and maternal demographics, were collected using a validated interviewer-administered questionnaire [24].

**Birth and child variables.**　Prematurity was defined as less than 37 weeks gestational age at birth. Gestational age was calculated based on a second trimester ultrasound; when this was unavailable maternal report of last menstrual period was used. Child growth measures were conducted at birth and all postnatal study visits. The birth anthropometric measurements were conducted by study staff. Infant length was measured in centimetres to the nearest completed 0.5 cm, where recumbent length was measured up until 18 months using a Seca length-measuring mat (Seca, Hamburg, Germany), and standing length was then measured at 24 months using a wall-mounted stadiometer. Weight was measured in kilograms to the nearest 10g, in light or no clothing, using the Tanita digital platform scale (TAN1584; IL, USA). Infant weight and length measurements were converted to z-scores that adjusted for infant age, sex, and prematurity; where the Fenton growth chart was applied to anthropometry measurements captured from premature children (< 37 weeks' gestation) at birth and up till 50 weeks postmenstrual age. WHO Anthro software was used to adjust for age and sex in measurements taken from full term infants at birth through to 2 years, as well as age-corrected measurements using the Fenton charts for premature infants [33]. Children were classified as ever stunted if they had a height-for-age z-score (HAZ) <-2 standard deviations [SD], underweight if they had a weight-for-age z-score (WAZ) <-2 SD, and wasted if they had a weight-for-height z-score (WHZ) <-2 SD.

## Statistical analysis

The primary outcome was all-cause hospitalisation. All hospitalisations from birth to two years of life were included. The child-time at risk in the cohort was calculated by starting calculations on the day after birth up to either two years of age, loss to follow up, or death, whichever occurred first. The time hospitalised was excluded from the at-risk period. Time at risk was censored by loss to follow up in the analyses. Hospital admissions due to maternal reasons were not included. Duration of hospitalisation was calculated from date of admission to date of discharge inclusive; where either date was missing, these hospital admissions were excluded from rate calculations but included in all other analyses (12; 3% of hospitalisations). We used an age-stratified approach to compare hospitalisations since prior literature highlights higher risks at early ages. We therefore categorised time periods into: (i) 0–12 months (infancy) with subdivisions of the first year into (ii) 0–6 months and (iii) 6–12 months, and (ii) 12–24 months. We did not sub-divide later ages due to smaller hospitalisation numbers in the second year. We report incidence rates for each of the time periods and investigated duration of hospitalisation. The main causes of hospitalisations are described and reported as a proportion of all hospitalisations (including those infants hospitalised at birth as a group), and separately as a proportion of hospitalisations subsequent to discharge after birth as a representation of community-acquired morbidity.

Results are presented for both HEU and HUU children as a whole and separately. Cox Proportional Hazards models were conducted to assess HIV exposure status differences in

hospitalisations, where standard errors were computed using the grouped jackknife method in order to account for recurrent hospitalisations. Crude incidence risk ratios (IRRs) were reported for unadjusted analyses, whilst adjusted hazard ratios (aHRs) were reported for multivariable analyses. Confounders were selected *a priori* using a directed acyclic directed acyclic graph (DAG) (S1 Fig). These included household income and maternal education. Separately we assessed any factors that significantly differed between HEU and HUU groups as potential confounders. Sensitivity analyses were conducted excluding recurrent hospitalisations.

Other potential risk and protective factors for hospitalisation including prematurity, breast-feeding, immunisation status, and HIV-related variables (maternal CD4, viral load and ART) were also assessed using hazard ratios. Finally, association of markers of malnutrition with all-cause hospitalisation rates in both HEU and HUU groups was investigated. All analyses were performed using R software.

### Ethics statement

The study was approved by the Human Research Ethics committee of the Faculty of Health Sciences University of Cape Town (RP 401/2009), and by the Provincial Health committee of the Western Cape's Health Department (RP 45/2011). Written informed consent was obtained from mothers at enrolment and was renewed annually.

## Results

### Sociodemographic characteristics

There were 1143 live births to 1137 mothers; of these, 2 [0.2%] children were diagnosed with HIV-infection and were excluded from this analysis. Five children died within one day of birth and were excluded as they never entered the at-risk period, leaving a total of 1136 children (247 [21.7%] HEU; 889 [78.3%] HUU). Cohort retention was high, with 1000 children in follow up at 2 years (87.5%; 1000/1143) with a total of 2075 child-year follow-up time. Loss to follow up was not associated with HIV exposure status ($\chi2 = 0.12$, p = 0.730). Baseline clinical and demographic characteristics of the cohort of 1136 mother-child dyads are provided in Table 1. One third of children were born to households with an income of <R1000 (~$75) per month (384; 34%), and almost two thirds of mothers had only primary school education (691; 61%). The median birthweight was 3.09 kg (IQR 2.71–3.42) and gestation was 39 weeks (IQR 38–40) at delivery; 186 (16%) children were born preterm, the majority (129; 69%) late preterm. Vaccination completion rates were high (995 children; 98% at 9 months), although 42% (409 children) experienced delayed vaccination at 9 months. HEU and HUU groups were comparable in household income, gestational age, birthweight and immunisation status, however, HEU children were born to mothers of older age with lower educational attainment. Across the cohort, 84% (896) children were ever breastfed, although median duration of exclusive breastfeeding was 1.5 months (IQR 0.7–3.2) and of any breastfeeding was 6 months (IQR 2–8); fewer HEU children were breastfed and for shorter duration compared to HUU children (p<0.001) (Table 1).

### Hospitalisation rates and mortality

Overall, 314 of 1136 children (27.6%) experienced a total of 430 hospitalisations; 23.6% (74/314) had >1 hospitalisation. Most hospitalisations occurred in the first year of life (incidence rate 314 (95% CI 281–349) per 1000-person-years, with the highest rates between 0–6 months [454 (95% CI 399–514) per 1000-person-years] and the lowest rates between 12–24 months [82 (95% CI 65–102) per 1000-person-years] (Table 2; S1 Table). Of the 1136 children, 18

**Table 1. Sociodemographic characteristics and comparison of HEU and HUU children and their mothers.**

| Variables | HEU (N = 247) | HUU (N = 889) | Total (N = 1136) | p-value[1] |
|---|---|---|---|---|
| *Maternal antenatal characteristics* | | | | |
| Household income (ZAR) | | | | 0.5 |
| < R1000 | 88 (36%) | 296 (33%) | 384 (34%) | |
| R1000-R5000 | 130 (53%) | 463 (52%) | 593 (52%) | |
| >R5000 | 29 (12%) | 129 (15%) | 158 (14%) | |
| Education | | | | <0.001 |
| Lower than secondary | 181 (73%) | 510 (57%) | 691 (61%) | |
| At least secondary or higher | 66 (27%) | 379 (43%) | 445 (39%) | |
| Employed | 62 (25%) | 245 (28%) | 307 (27%) | 0.5 |
| Marital status[2] (Married/cohabitating) | 106 (43%) | 351 (39%) | 457 (40%) | 0.4 |
| Alcohol use in pregnancy[2] | 23 (9.7%) | 112 (14%) | 135 (13%) | 0.2 |
| Smoking in pregnancy[2] | 30 (12%) | 290 (33%) | 320 (28%) | <0.001 |
| Age at delivery, median (IQR) years | 30 (26,34) | 25 (22,30) | 26 (22,31) | <0.001 |
| *Infant characteristics* | | | | |
| Sex (Male) | 137 (55%) | 444 (50%) | 581 (51%) | 0.14 |
| Gestational age at delivery, median (IQR) weeks | 39 (37,40) | 39 (38,40) | 39 (38,40) | 0.2 |
| Preterm (<37 weeks' gestation delivery) | 47 (19%) | 139 (16%) | 186 (16%) | 0.2 |
| Birthweight (kg), median (IQR)[2] | 3.08 (2.66,3.36) | 3.10 (2.72,3.43) | 3.09 (2.71,3.42) | 0.4 |
| Low birthweight (< 2500g)[2] | 35 (14%) | 135 (15%) | 170 (15%) | 0.8 |
| Vaccination completed[2] - | | | | |
| 14 weeks | 226 (100%) | 779 (99%) | 1005 (99%) | >0.9 |
| 9 months | 217 (98%) | 738 (99%) | 955 (98%) | >0.9 |
| Vaccination timing[2] | | | | |
| 14 weeks: Delayed | 63 (28%) | 229 (29%) | 292 (29%) | 0.8 |
| 9 months: Delayed | 91 (41%) | 318 (42%) | 409 (42%) | >0.9 |
| Rotavirus vaccination completed | | | | |
| 6-10 weeks | 231 (100%) | 804 (99%) | 1035 (99%) | >0.9 |
| 14 weeks | 226 (100%) | 770 (98%) | 996 (99%) | 0.3 |
| Rotavirus vaccination timing | | | | |
| 6-10 weeks: Delayed | 15 (6.5%) | 61 (7.6%) | 76 (7.4%) | 0.7 |
| 14 weeks: Delayed | 61 (27%) | 207 (27%) | 268 (27%) | >0.9 |
| Ever breastfed[2] | 107 (46%) | 789 (95%) | 896 (84%) | <0.001 |
| Exclusively breastfed (months), median (IQR)[2] | 0 (0,2.8) | 1.8 (0.9,3.2) | 1.5 (0.7, 3.2) | <0.001 |
| Exclusive breastfeeding (6 months)[2] | 30 (13%) | 91 (11%) | 121 (11%) | 0.5 |
| Any breastfeeding (months), median (IQR)[2] | 0 (0,6) | 9 (3,24) | 6 (2, 18) | <0.001 |
| Breastfeeding of one year [2] | 26 (11%) | 374 (45%) | 400 (38%) | <0.001 |
| *Maternal antenatal HIV-related variables* | | | | |
| CD4, mean (SD)[2] | 457 (241) | - | - | - |
| CD4[2] | | | | |
| <=500 | 124 (62%) | - | - | - |
| >500 | 75 (38%) | | | |
| Viral load[2] | | | | |
| <40 copies/ml (undetectable) | 93 (64%) | - | - | |
| ≥40 copies/ml | 52 (36%) | | | |
| ART regimen initiation[2] | | | | |
| Before pregnancy | 96 (40%) | - | - | |

*(Continued)*

**Table 1.** (Continued)

| Variables | HEU (N = 247) | HUU (N = 889) | Total (N = 1136) | p-value[1] |
|---|---|---|---|---|
| During pregnancy | 144 (60%) | - | - | |
| ART regimen[2] | | | | |
| PMTCT/AZT monotherapy | 36 (15%) | - | - | |
| First line (NNRTI + 2NRTIs) | 191 (80%) | - | - | |
| Second line (PI-containing) | 13 (5.4%) | - | - | |

*Footnote*: 1 = Inferential tests stratified by HIV exposure status. Chi-squared tests used for categorical variables & Mann Whitney U test used for continuous variables. Percentages are given out of available data; missing data: household income & smoking n = 1; Alcohol use = 73; birthweight n = 3; vaccinations at 14 weeks n = 125, at 9 months n = 166; rotavirus at 6–10 weeks n = 94, at 14 weeks n = 126; breastfeeding n = 71; CD4 n = 48; viral load n = 102; ART n = 7. Vaccination delay defined as any of the vaccinations given at 14 weeks (or 9 months) and younger delayed by more than 2 weeks of due date. Abbreviations: ART = antiretroviral therapy; AZT = zidovudine; HEU = HIV-exposed uninfected; HUU = HIV-unexposed uninfected; NRTI = nucleoside reverse transcriptase inhibitor; NNRTI = non-nucleoside reverse transcriptase inhibitor; ZAR = South African rand

infants died (1.58%) in the first year of life (HEU 7/247 [2.83%]; HUU 11/889 [1.24%] $\chi^2$ = 2.22, p = 0.136).

**Causes of hospitalisation.** Overall, LRTI was the most common cause of hospitalisation (39.3% of cases; 169/430), with gastroenteritis (17.7%; 76/430) second, across all ages and in the HEU and HUU groups respectively (Table 3). However, hospitalisations due to gastroenteritis were proportionally higher in HEU versus HUU groups in infancy (OR = 2.35 [95% CI 1.36–4.04], p<0.01) (Table 3; S2 Fig), while other causes did not differ between groups. Overall, 20% (84/430) of hospitalisations occurred during the birth admission period. Of hospitalisations subsequent to discharge after birth (346), 83% (288/346) had infectious causes; LRTIs caused 49% (169/346), with a high proportion (31%; 53/169) due to RSV. In the first six months of life (164 hospitalisations), RSV-LRTI was responsible for 22% of all hospitalisations (36/164) and caused 41% (36/88) of LRTI hospitalisations (Table 3; S2 Table). While gastroenteritis caused 22% (76/346) of admissions after birth, other infections were responsible for 12% (43/346) of community-acquired hospitalisations. These included meningitis, sepsis, otitis media, urinary tract infection, or periorbital cellulitis. Finally, other causes including injuries,

**Table 2. Rate of hospitalisations in the first two years by HIV exposure status.**

| | | Unadjusted | | | Adjusted model 1 | Adjusted model 2 | Adjusted model 3 |
|---|---|---|---|---|---|---|---|
| | All IR / 1000 person years (95% CI) | HEU IR / 1000 person years (95% CI) | HUU IR /1000 person years (95% CI) | IRR (95% CI) | HR (95% CI) | HR (95% CI) | HR (95% CI) |
| | *All hospitalisations* | | | | | | |
| 0 – ≤12 months | 314 (281-349) | 450 (368-545) | 276 (242-314) | 1.63 (1.29-2.05) *** | 1.47 (1.15-1.87) ** | 1.54 (1.19-2.00) ** | 1.46 (1.13-1.89) ** |
| 0 – <6 months | 454 (399-514) | 627 (492-787) | 406 (348-471) | 1.54 (1.18-2.03) ** | 1.43 (1.09-1.87) ** | 1.53 (1.14-2.05) ** | 1.40 (1.05-1.86) * |
| 6 – ≤12 months | 168 (135-207) | 265 (179-379) | 142 (107-183) | 1.87 (1.20-2.91) ** | 1.73 (1.05-2.85) * | 1.76 (1.00-3.08) * | 1.76 (1.05-2.94) * |
| 12 – ≤24 months | 82 (65-102) | 77 (45-124) | 84 (65-106) | 0.92 (0.54-1.57) | 0.91 (0.53-1.55) | 0.78 (0.44-1.38) | 0.91 (0.53-1.56) |
| 0 – ≤24 months | 269 (247-291) | 268 (222-320) | 183 (163-205) | 1.47 (1.19-181) *** | 1.34 (1.07-1.67) * | 1.36 (1.07-1.72) * | 1.32 (1.04-1.67) * |

*Footnote*: Unadjusted incident rate ratios and adjusted hazard models for hospitalisations in the first 2 years of life by HIV exposure. Multivariate models adjusted for (1) maternal education and household income; (2) maternal education, household income, maternal age at birth and maternal smoking; (3) maternal education, household income, exclusive breastfeeding duration *** p-value < 0.001; ** p-value < 0.01; * p-value < 0.05 Abbreviations: OR = Odds ratio; IR = Incidence rate; IRR = Incidence rate ratio; HR = Hazard ratio; HEU = HIV-exposed uninfected; HUU = HIV-unexposed uninfected

**Table 3. Causes of hospitalisation stratified by age and HIV exposure status.**

| Cause | Total | | | HEU | | | HUU | | | HEU versus HU |
|---|---|---|---|---|---|---|---|---|---|---|
| | N | % all hosp. (n=430) | % hosp. excluding birth‡(n=346) | N | % all hosp. (n=124) | % hosp. excluding birth (n=100) | N | % all hosp. (n=306) | % hosp. excluding birth(n=246) | OR (95% CI)[1] |
| **LRTI#** | | | | | | | | | | |
| 0-12 months | 131 | 30% | 38% | 38 | 31% | 38% | 93 | 30% | 38% | 0.87 (0.54-1.39) |
| 0-6 months | 88 | 20% | 25% | 26 | 21% | 26% | 62 | 20% | 25% | 1.01 (0.57-1.78) |
| 6-12 months | 43 | 10% | 12% | 12 | 10% | 12% | 31 | 10% | 13% | 0.56 (0.23-1.34) |
| 12-24 months | 38 | 9% | 11% | 7 | 6% | 7% | 31 | 10% | 13% | 0.78 (0.26-2.18) |
| **RSV-LRTI** | | | | | | | | | | |
| 0-12 months | 46 | 11% | 13% | 8 | 6% | 8% | 38 | 12% | 15% | 0.42 (0.16-1.00) |
| 0-6 months | 36 | 8% | 10% | 6 | 5% | 6% | 30 | 10% | 12% | 0.36 (0.12-1.01) |
| 6-12 months | 10 | 2% | 3% | 2 | 2% | 2% | 8 | 3% | 3% | 0.57 (0.08-2.83) |
| 12-24 months | 7 | 2% | 2% | 1 | 1% | 1% | 5 | 2% | 2% | 0.92 (0.04-7.64) |
| **Gastroenteritis** | | | | | | | | | | |
| 0-12 months | 53 | 12% | 15% | 26 | 21% | 26% | 27 | 9% | 11% | 2.35 (1.36-4.04) ** |
| 0-6 months | 25 | 6% | 7% | 12 | 10% | 12% | 13 | 4% | 5% | 2.45 (1.16-5.14) * |
| 6-12 months | 28 | 7% | 8% | 14 | 11% | 14% | 14 | 5% | 6% | 2.16 (0.90-5.27) |
| 12-24 months | 23 | 5% | 7% | 6 | 5% | 6% | 17 | 6% | 7% | 1.72 (0.61-4.86) |
| **Other infections** | | | | | | | | | | |
| 0-12 months | 31 | 7% | 9% | 7 | 6% | 7% | 24 | 8% | 10% | 0.77 (0.48-1.24) |
| 0-6 months | 18 | 4% | 5% | 4 | 3% | 4% | 14 | 5% | 6% | 0.65 (0.35-1.18) |
| 6-12 months | 13 | 3% | 4% | 3 | 2% | 3% | 10 | 3% | 4% | 0.93 (0.39- 2.21) |
| 12-24 months | 12 | 3% | 3% | 1 | 1% | 1% | 11 | 4% | 4% | 0.65 (0.22-1.82) |
| **Other causes †** | | | | | | | | | | |
| 0-12 months | 41 | 10% | 12% | 10 | 8% | 10% | 31 | 10% | 13% | 0.66 (0.38-1.14) |
| 0-6 months | 33 | 8% | 10% | 7 | 6% | 7% | 26 | 8% | 11% | 0.71 (0.38-1.27) |
| 6-12 months | 8 | 2% | 2% | 3 | 2% | 3% | 5 | 2% | 2% | 0.60 (0.08-2.79) |
| 12-24 months | 17 | 4% | 5% | 5 | 4% | 5% | 12 | 4% | 5% | 1.14 (0.24-4.28) |
| **Birth hospitalisations** | | | | | | | | | | |
| Neonatal | 84 | 20% | - | 24 | 19% | - | 60 | 20% | - | 0.86 (0.50-1.47) |

*Footnote*: † Other causes include accident and trauma, burns, seizures, nutritional issues (failure to thrive, protein energy malnutrition, anaemia)

‡Hospitalisations subsequent to discharge after birth

#Inclusive of RSV-LRTI

** p-value < 0.01

* p-value < 0.05

Abbreviations: HEU = HIV-exposed uninfected; HUU = HIV-unexposed uninfected; Hosp = Hospitalisation; LRTI = Lower respiratory tract infection;

RSV = Respiratory Syncytial Virus; 1 = HEU relative to HUU group

trauma, burns, seizures, jaundice or nutritional issues (failure to thrive, malnutrition, or anaemia), comprised 17% (58/346) of hospitalisations.

## Hospitalisation by HIV exposure status

Children who were HEU had a significantly greater probability of hospitalisation (Fig 1). Compared to HUU children, the incidence rate of hospitalisation was significantly higher for HEU children from 0 to 12 months (IRR 1.63 [95% CI 1.29–2.05], and this held across both the 0–6 month (IRR 1.54 [95% CI 1.18 to 2.03]) and 6–12 month age stratifications (1.87 [95% CI 1.20 to 2.91]) (Table 2); there was no significant difference from 12–24 months (0.92 [95% CI 0.54–

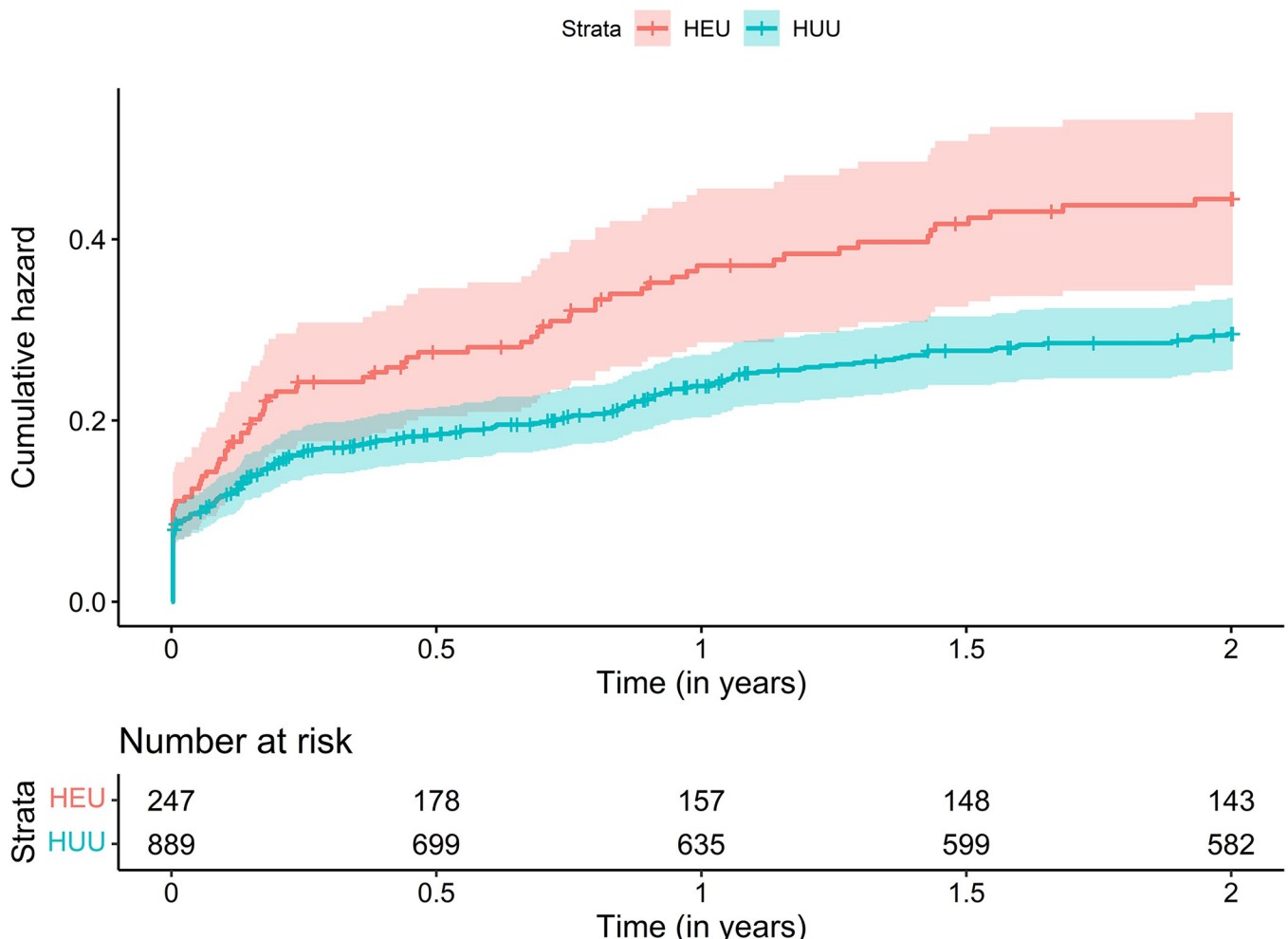

**Fig 1. Cumulative hazard function of hospitalisations by HIV exposure status.** Cumulative hazard function calculated for first episode hospitalisations.

1.57]). Findings were consistent between unadjusted and adjusted models including maternal education, household income, maternal smoking in pregnancy, and breastfeeding covariates (Table 2). Separately excluding recurrent hospitalisations or birth hospitalisations marginally impacted the estimate sizes and confidence intervals, although results were similar (S3 and S4 Tables respectively). Children who were HEU had significantly longer hospitalisation admissions compared to HUU children (mean [SD] = 5.8 [6.9] vs 3.6 [4.1] days; p = 0.0039) (Fig 2).

## Other determinants of hospitalisation

Prematurity was associated with increased likelihood of hospitalisation in all infants (HR 2.82 [95% CI 2.28–3.49] (Table 4), with the greatest effect seen in the first 6 months of life. On stratifying by HIV exposure, prematurity continued to be associated with increased risk of hospitalisation at 6–12 months in HEU children (2.66 [95% CI 1.07–6.62]). Ever breastfeeding (HR 0.69 [0.53–0.90]) protected against hospitalisation in infancy, and breastfeeding for a year had a similar effect (HR 0.67 [95% CI 0.51–0.89]) (Table 4). Additionally, exclusive breastfeeding duration and breastfeeding for the first year were associated with reduced risk of gastroenteritis specifically (OR 0.83 [95% CI 0.71–0.96], p-value = 0.019; OR 0.92 [95% CI 0.88–0.96], p-

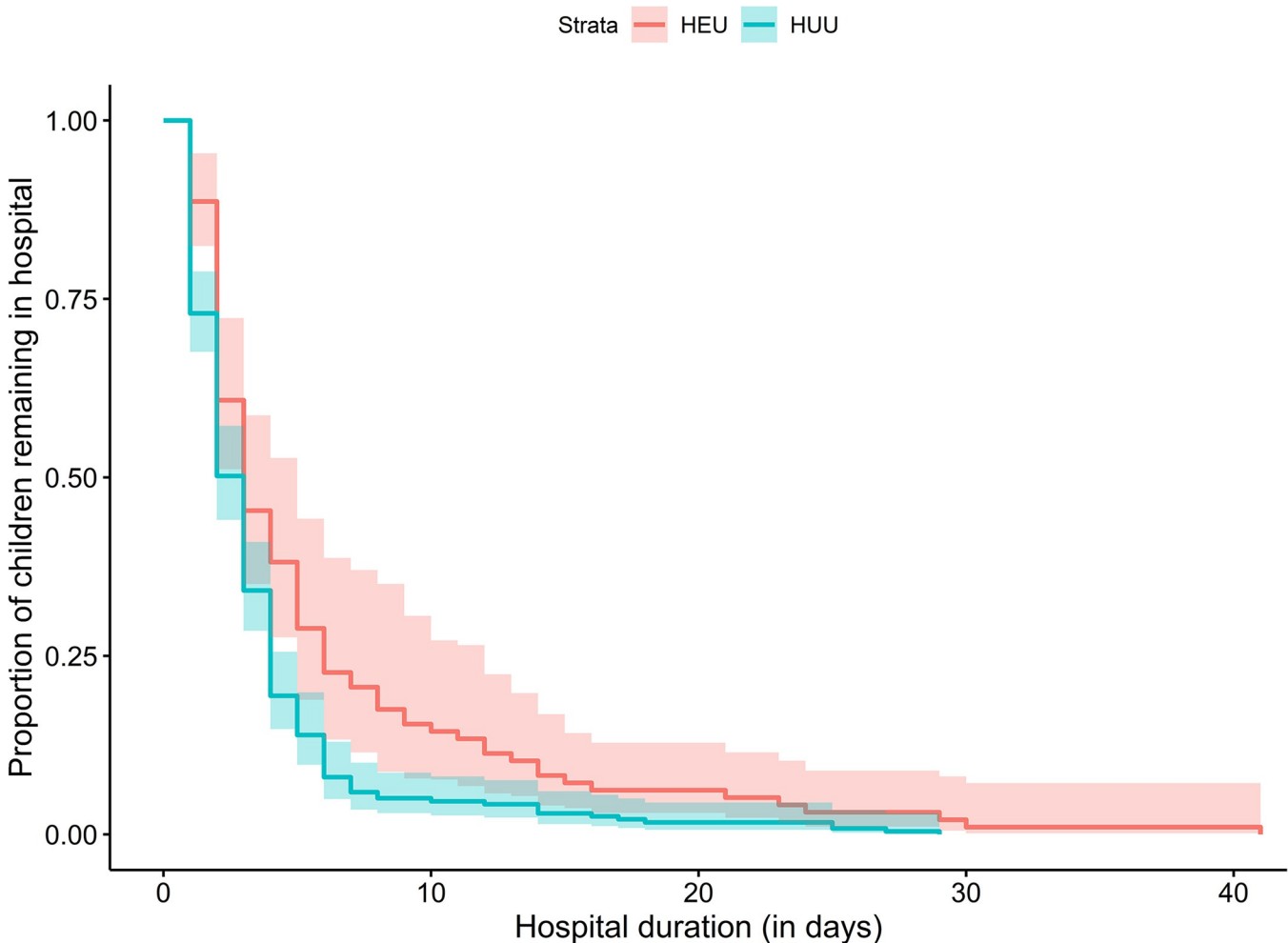

**Fig 2. Comparison of duration of hospitalisation by HIV exposure status.** Comparison of duration of hospitalisation between children who were HEU and HUU, *p* = 0.0039. Duration excluded birth hospitalisations.

value <0.001 respectively). Delay in vaccinations at 9 months was associated with increased hospitalisations between 0–12 months (HR 1.43 [95% CI 1.12–1.82]), only for LRTI hospitalisation (OR = 2.01 [95% CI 1.25–3.25], p = 0.004). When birth hospitalisations were excluded, similar associations were found between prematurity, breastfeeding and vaccination delay and hospitalisation (S5 Table), although hazard ratios were lower for prematurity associations.

Of maternal HIV-related factors, a detectable maternal HIV viral load during pregnancy (>40 copies/ml) was associated with hospitalisation in the HEU group between 6–12 months compared to mothers living with HIV who had an undetectable viral load (HR 4.43 [95% CI 1.52–12.87]) but there were no significant associations with maternal CD4 or timing of ART initiation.

In terms of concomitant malnutrition, there was a trend for the HEU group of hospitalized children to be more underweight, stunted, and/or wasted, across all hospitalisations compared to the hospitalised HUU group, particularly in the 12–24 month period. However, this did not reach statistical significance (S6 Table).

**Table 4. Impact of birth, breastfeeding and HIV-related factors on hospitalisations stratified by HIV exposure status and age.**

| | HEU HR (95% CI) | HUU HR (95% CI) | Total HR (95% CI) |
|---|---|---|---|
| *Birth factors* | | | |
| **Prematurity** | | | |
| 0-12 months | 2.92 (2.02-4.22) *** | 2.71 (2.09-3.51) *** | 2.82 (2.28-3.49) *** |
| 0-6 months | 3.19 (2.14-4.76) *** | 3.34 (2.52-4.43) *** | 3.45 (2.66-4.22) *** |
| 6-12 months | 2.66 (1.07-6.62) * | 1.26 (0.66-2.41) | 1.74 (0.99-3.04) |
| 12-24 months | 1.48 (0.43-5.11) | 0.97 (0.45-2.05) | 1.07 (0.56-2.04) |
| *Feeding* | | | |
| **Ever breastfed** | | | |
| 0-12 months | 0.92 (0.59-1.42) | 0.80 (0.48-1.32) | 0.69 (0.53-0.90) ** |
| 0-6 months | 0.78 (0.48-1.27) | 0.81 (0.46-1.45) | 0.67 (0.49-0.90) ** |
| 6-12 months | 1.30 (0.56-3.01) | 0.69 (0.26-1.84) | 0.71 (0.40-1.26) |
| 12-24 months | 1.01 (0.39-2.62) | 1.07 (0.35- 3.22) | 1.08 (0.60- 1.94) |
| **Breastfeeding for a year** | | | |
| 0-12 months | 1.29 (0.66-2.51) | 0.67 (0.49-0.91) ** | 0.67 (0.51-0.89) ** |
| 12-24 months | 0.97 (0.24-3.88) | 0.64 (0.36-1.11) | 0.69 (0.42-1.16) |
| *Vaccination* | | | |
| **Timing by 9 months** | | | |
| 0-12 months | 1.43 (0.92-2.22) | 1.43 (1.07-1.91) * | 1.43 (1.12-1.82) ** |
| 12-24 months | 1.34 (0.52-3.48) | 1.03 (0.60-1.77) | 1.09 (0.68-1.75) |
| *HIV-related variables* | | | |
| **Maternal CD4** | | | |
| *(>500 vs ≤500)* | | | |
| 0-12 months | 0.99 (0.61-1.58) | - | - |
| 0-6 months | 0.94 (0.56-1.59) | - | - |
| 6-12 months | 1.11 (0.43-2.83) | - | - |
| 12-24 months | 0.88 (0.33-2.35) | - | - |
| **Maternal Viral load** | | | |
| *(≥ 40 vs <40)* | | | |
| 0-12 months | 1.40 (0.85-2.31) | - | - |
| 0-6 months | 1.03 (0.58-1.82) | - | - |
| 6-12 months | 4.43 (1.52-12.87) ** | - | - |
| 12-24 months | 1.31 (0.40-4.30) | - | - |
| **ART regimen initiation** | | | |
| *(Before vs during pregnancy)* | | | |
| 0-12 months | 0.83 (0.55-1.26) | - | - |
| 0-6 months | 0.94 (0.59-1.48) | - | - |
| 6-12 months | 0.54 (0.20-1.40) | - | - |
| 12-24 months | 0.76 (0.29-1.99) | - | - |

Footnote

*** p-value < 0.001

** p-value < 0.01

* p-value < 0.05 for unadjusted models.

Definitions: Prematurity (<37 vs > = 37 weeks); Ever breastfeeding (>0 months vs = 0 months); Breastfeeding for a year (classified as >11 months vs < = 11 months). Abbreviations: ART = antiretroviral therapy; HEU = HIV-exposed uninfected; HUU = HIV-unexposed uninfected; HR = Hazard ratio

## Discussion

In this population-based birth cohort we found high rates of early-life hospitalisation and that more than a quarter of children experienced hospital admission, representing a substantial burden. Secondly, LRTI was the most common reason for hospitalisation overall, largely driven by RSV-LRTI. Thirdly, HIV exposure was associated with a greater risk of hospitalisation in infancy and with longer duration of admission. Finally, prematurity and delayed vaccination were drivers of hospitalisation for all children, and higher maternal HIV viral load in HEU children, while breastfeeding was protective, particularly from gastroenteritis.

Overall, the incidence of hospitalisation was high, despite over 98% immunization coverage and excluding child HIV infection, ranging from 454/1000 person years in the first 6 months to 82/1000 person years between 12–24 months. This is consistent with previous reported hospitalisation rates in studies from South Africa [34]. The incidence of hospitalisation was greatest between 0–6 months of age, declining substantially into the second year, highlighting the vulnerability and importance of the first 6 months of life, and the potential for maternal immunisation as a preventive strategy.

Infections were the predominant cause of hospitalisation, and our results support LRTI and diarrhoea as key causes of morbidity in children [22]. LRTI was the commonest reason for hospital admission, responsible for almost half of all hospitalisations outside of those occurring at birth. Additionally, in the first 6 months of life RSV-LRTI was responsible for 22% of all-cause hospitalisations and for 41% of LRTI hospitalisations. These data are consistent with our prior finding that RSV is the leading cause of LRTI in this setting [35], especially in infants [27, 28], and with global data of the importance of RSV-LRTI in LMICs [36]. Importantly these findings highlight the potential impact of new RSV preventive interventions (either monoclonal antibody or maternal vaccination), to reduce LRTI and associated hospitalisations in LMIC settings [37, 38].

Our findings add to data from other studies that suggest an impact of HIV exposure on hospitalisation incidence and duration in infants, in both LMIC [20, 21] and high income countries [39], despite the scale up of maternal ART. However, our results show a longer at-risk period, until 12 months, compared to another South African study, which found infection-related hospitalisation risk lasted up to 3 months of age [20]. The increased hospitalisation risk may be due to multiple factors associated with early-life infection [14], including higher exposure to opportunistic infections in HIV-households, immune dysregulation or reduced transplacental antibody transfer, and low breastfeeding rates. Studies have found that children who are HEU have lower transplacental transfer of maternal antibodies to specific pathogens and some infants have altered responses to vaccination [40, 41]. However, another USA study reported HEU infants have increased hospitalisation despite robust antibody responses to vaccines [42], suggesting cellular immune defects may underlie the susceptibility to infection. Detectable maternal HIV viral load in the peripartum period was a risk factor for HEU child hospitalisation, as reported by others [20, 43]. This is an indicator of maternal disease severity and immune compromise and suggests viral load monitoring and optimising ART adherence as strategies for prevention. We did not show an effect of timing of ART initiation (prior to pregnancy versus during pregnancy) which differs from a European study, where initiation of maternal ART prior to pregnancy reduced child hospitalisation from infectious causes [44].

In addition to HIV exposure, we identified several other factors associated with hospitalisation. Prematurity was a risk factor for hospitalisation for all children adding to evidence of the association of prematurity with hospitalisation due to infectious causes [45]. Further, in HEU children, prematurity was more strongly associated with hospitalisation compared to HUU children, despite no HEU/HUU group differences in preterm birth or birthweight, suggesting

an additive impact of multiple risk factors. Another study also found prematurity contributed to increased infection-related hospitalisation in HEU children [21], however, in this study timely infant vaccination was poor and there were more preterm and low birthweight HEU than HUU children. Overall vaccination completion in our cohort was very high. However, vaccination delay was associated with hospitalisation, particularly for LRTI, consistent with evidence that optimising vaccination reduces hospitalisation risk [20]. These data highlight the importance of timely vaccination delivery [22], an issue that has become even more important during the COVID-19 pandemic which has impacted on childhood immunisation programs.

Breastfeeding was protective against hospitalisation, although we had reduced power to detect an effect due to low breastfeeding rates across the cohort. WHO recommends exclusive breastfeeding from birth until six months of age, with continued breastfeeding alongside appropriate complementary foods thereafter, however, only 13% of the cohort breastfed to six months. Prior to 2013, breastfeeding was not recommended for women living with HIV which may have affected feeding practices and reduced our power to show an effect in HEU children. Our results identifying breastfeeding as a protective factor are consistent with substantial evidence demonstrating the importance of optimal breastfeeding in preventing morbidity [20, 46]. However, in the comparison of HEU and HUU groups, our hazard ratios held on adjusting for breastfeeding, fitting with other studies that have found breastfeeding reduces, but does not completely eliminate, morbidity in children who are HEU [47–49]. We found a strong association between breastfeeding and reduced hospitalisation for gastroenteritis, which may explain the higher rates of gastroenteritis-associated hospitalisation in HEU children from 0–6 months. Both LRTI and gastroenteritis have previously been reported to be higher in HEU children than HUU [12, 18, 39, 50–52], and early, exclusive, and prolonged breastfeeding are known to be protective [43].

Strengths of this study include the well-characterised cohort with detailed prospective data collection, high vaccination coverage, no paediatric HIV, and high cohort retention in a low socioeconomic status African setting. The longitudinal design with comparable controls allowed the assessment of HIV exposure status with heterogeneity in antiretroviral combinations and maternal immune status across the cohort which is representative of child populations in other LMIC settings across sub-Saharan Africa. Limitations include the potential for missing hospitalisations; however, all children were hospitalised at a single central hospital as this is the only public hospital serving the area, active surveillance was carefully done, and there were low rates of loss to follow up across the cohort. While bias could have been introduced into the analyses from loss to follow up, we attempted to minimise the impact through stratifying by age and by implementing measures to reduce loss to follow up and ensure all hospitalisations were captured. The prevalence of loss to follow up was low and we confirmed this was not associated with HIV exposure status. The low breastfeeding rates across the cohort may have reduced power to detect differences, and the study procedures need to be replicated in populations with high breastfeeding rates. Finally, the results may not be generalisable to high malaria prevalence settings or settings of high child HIV infection.

## Conclusion

Despite high vaccination rates and no child HIV infection, there was a high incidence of hospitalisation, especially during infancy in HEU children, also associated with prematurity, delayed vaccinations, and lack of breastfeeding. LRTI was the most common reason for hospitalisation, and RSV was a predominant pathogen. Our findings indicate the potential for simple, affordable interventions to mitigate hospitalisation risk including encouraging prolonged breastfeeding and timely vaccination as per global guidance [53]. We also demonstrate the importance of

targeting caregivers and families of at-risk groups, including those who are premature and HIV-exposed, with prevention and intervention strategies [22]. In children who are HEU, optimising antenatal maternal ART and monitoring viral load may decrease morbidity. Finally, the high prevalence of RSV-LRTI as a contributor to hospitalisation especially in the first 6 months of life, suggest that new interventions to prevent RSV may have an impact in reducing hospitalisation.

## Supporting information

**S1 Fig. DAG model.**
(PDF)

**S2 Fig. Causes of hospitalisation stratified by HIV exposure and age groups.**
(PDF)

**S1 Table. Incidence of hospitalisation in the first two years of life stratified by age.**
(PDF)

**S2 Table. Proportion of LRTI and RSV-LRTI hospitalisations by age category.**
(PDF)

**S3 Table. Incidence of hospitalisations in the first two years of life by HIV exposure status excluding recurrent events.**
(PDF)

**S4 Table. Incidence of hospitalisations in the first two years of life by HIV exposure status excluding birth hospitalisations.**
(PDF)

**S5 Table. Impact of birth, breastfeeding, and HIV-related factors on hospitalisation, stratified by HIV exposure status and age excluding birth hospitalisations.**
(PDF)

**S6 Table. Association between malnutrition and hospitalisation.**
(PDF)

## Acknowledgments

We thank the study staff, the clinical and administrative staff of the Western Cape Government Health Department at Paarl Regional Hospital, Mbekweni and TC Newman clinics for their support of the study. We are extremely grateful to the families and children who participated in this study. We thank our collaborators and postgraduate students for work on the study.

## Author Contributions

**Conceptualization:** Catherine J. Wedderburn, Heather J. Zar.

**Data curation:** Julia Bondar, Marilyn T. Lake, Raymond Nhapi, Whitney Barnett, Mark P. Nicol.

**Formal analysis:** Julia Bondar, Marilyn T. Lake, Raymond Nhapi.

**Funding acquisition:** Heather J. Zar.

**Investigation:** Catherine J. Wedderburn, Whitney Barnett, Mark P. Nicol, Heather J. Zar.

**Methodology:** Catherine J. Wedderburn, Julia Bondar, Marilyn T. Lake, Whitney Barnett, Mark P. Nicol, Liz Goddard, Heather J. Zar.

**Project administration:** Whitney Barnett.

**Resources:** Heather J. Zar.

**Validation:** Catherine J. Wedderburn, Julia Bondar, Marilyn T. Lake, Raymond Nhapi.

**Visualization:** Marilyn T. Lake.

**Writing – original draft:** Catherine J. Wedderburn.

**Writing – review & editing:** Catherine J. Wedderburn, Julia Bondar, Marilyn T. Lake, Raymond Nhapi, Whitney Barnett, Mark P. Nicol, Liz Goddard, Heather J. Zar.

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
