## [Decision Letter · Decision Letter 0]

8 Oct 2023

PGPH-D-23-01503

Risk and rates of hospitalisation in young children: a prospective study of a South African birth cohort

Dear Dr. Wedderburn,

Thank you for submitting your manuscript to PLOS Global Public Health. After careful consideration, we feel that it has merit but does not fully meet PLOS Global Public Health’s publication criteria as it currently stands. Therefore, we invite you to submit a revised version of the manuscript that addresses the points raised during the review process.

We look forward to receiving your revised manuscript.

Kind regards,

Abram L. Wagner, PhD, MPH

Academic Editor

Journal Requirements:

2. Please send a completed 'Competing Interests' statement, including any COIs declared by your co-authors. If you have no competing interests to declare, please state "The authors have declared that no competing interests exist". Otherwise please declare all competing interests beginning with twhe statement "I have read the journal's policy and the authors of this manuscript have the following competing interests:"

3. Please provide separate figure files in .tif or .eps format only and remove any figures embedded in your manuscript file. Please also ensure all files are under our size limit of 10MB.

4. In the online submission form, you indicated that "The de-identified data that support the findings of this study are available from the authors upon reasonable request as per DCHS cohort guidelines". All PLOS journals now require all data underlying the findings described in their manuscript to be freely available to other researchers, either 1. In a public repository, 2. Within the manuscript itself, or 3. Uploaded as supplementary information.

Additional Editor Comments (if provided):

Note that some comments are in PDF attachments available in Editorial Manager.

Reviewers' comments:

Reviewer's Responses to Questions

**Comments to the Author**

1. Does this manuscript meet PLOS Global Public Health’s publication criteria? Is the manuscript technically sound, and do the data support the conclusions? The manuscript must describe methodologically and ethically rigorous research with conclusions that are appropriately drawn based on the data presented.

Reviewer #1: Yes

Reviewer #2: Yes

Reviewer #3: Yes

Reviewer #4: Yes

Reviewer #5: Yes

2. Has the statistical analysis been performed appropriately and rigorously?

Reviewer #1: Yes

Reviewer #2: Yes

Reviewer #3: Yes

Reviewer #4: Yes

Reviewer #5: Yes

3. Have the authors made all data underlying the findings in their manuscript fully available (please refer to the Data Availability Statement at the start of the manuscript PDF file)?

Reviewer #1: Yes

Reviewer #2: Yes

Reviewer #3: No

Reviewer #4: Yes

Reviewer #5: Yes

4. Is the manuscript presented in an intelligible fashion and written in standard English?

Reviewer #1: Yes

Reviewer #2: Yes

Reviewer #3: Yes

Reviewer #4: Yes

Reviewer #5: No

5. Review Comments to the Author

Reviewer #1: This is a very insightful article highlighting the relevance of child survival strategies like immunization and breastfeeding to the reduction of under 5 morbidity and mortality. Moreso, this was done among HIV exposed and unexposed children which is informative as far as PMTCT is concerned. More needs to be done among the HIV mothers particularly as regards the benefits of breastfeeding in LMIC.

Kindly check the section of statistical analysis, second paragraph, line 3, "jack knife method in order account for recurrent" Add "to" after "order"

Reviewer #2: in the discussion section (page 11), authors may wish to replace the word "ameliorate" with "eliminate"

However, in the comparison of HEU and HUU groups, our hazard ratios held on adjusting for breastfeeding, fitting with other studies that have found breastfeeding reduces, but does not completely eliminate, morbidity in children who are HEU.

Reviewer #3: This is an excellent study and most areas are covered adequately.

However, additional information may be useful in the following sections.

Abstract: Besides the explanation, specifically state the study design used.

Data availability statement: The given response sounds restrictive and depends on authors' willingness to avail the same. Nonetheless, an undertaking is required to avail data in a public repository or similar and indicate the link.

Methods: Subheading study population - use correct name of the single hospital involved as Paarl Regional Hospital and add more information about the setting eg catchment population and surrounding clinics; Subheading hospitalizations - were all treating clinicians trained on WHO clinical case definition of LRTI? Also, all children with LRTI had a nasopharyngeal swab taken to look for RSV. That practice is not routine care in SA public health sector. How was the RSV tests funded? Subheading birth and child variables: check and reference definition of prematurity because it doesn't include 37 weeks.

Results: Socio-demographics section appears lengthy. Its recommended that authors focus on 7 significant variables here and show the rest in appendix (as seen in Table I).

Hospitalization rates and mortality p7. Review and re-write this equation: HUU 11 [1.24%] χ2 = 2.22, p = 0.136).

Statistics: Can authors explain how loss to follow-up was addressed

Discussion: How were potential biases handled? And what did the authors do to minimize limitations on study findings?

Conclusion: Alright

Reviewer #4: In conclusion, considering the alignment with the journal's focus, methodological rigor, clear presentation, relevance, ethical considerations, and potential for improvement, the article meets the criteria for publication in the journal.

Reviewer #5: I thank the authors for a generally well written manuscript. Subject to addressing the various issues i have observed and especially, change of title to what I have suggested in the reviewed PDF or giving a clear reason for exclusion of HIV exposed infected (HEI) children in analysis for the suggested title by the authors to be maintained; the manuscript may be published.

6. PLOS authors have the option to publish the peer review history of their article (what does this mean?). If published, this will include your full peer review and any attached files.

**Do you want your identity to be public for this peer review?** For information about this choice, including consent withdrawal, please see our Privacy Policy.

Reviewer #1: **Yes: **Suleiman E. Mshelia

Reviewer #2: No

Reviewer #3: **Yes: **John Mukuka Musonda

Reviewer #4: No

Reviewer #5: **Yes: **Benedicto Mugabi

---

## [Editor Report · Decision Letter 1]

5 Dec 2023

Risk and rates of hospitalisation in young children: a prospective study of a South African birth cohort

PGPH-D-23-01503R1

Dear Dr Wedderburn,

We are pleased to inform you that your manuscript 'Risk and rates of hospitalisation in young children: a prospective study of a South African birth cohort' has been provisionally accepted for publication in PLOS Global Public Health.

Best regards,

Abram L. Wagner, PhD, MPH

Academic Editor